# The Hydrodynamics of a Micro-Rocket Propelled by a Deformable Bubble

**Giacomo Gallino [1], Lailai Zhu [1,2,3] 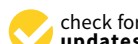 and François Gallaire [1,***

[1]  Laboratory of Fluid Mechanics and Instabilities, EPFL, CH-1015 Lausanne, Switzerland;
    giaco5988@gmail.com (G.G.); lailaizhu00@gmail.com (L.Z.)
[2]  Linné Flow Centre and Swedish e-Science Research Centre (SeRC), KTH Mechanics,
    SE 10044 Stockholm, Sweden
[3]  Department of Mechanical and Aerospace Engineering, Princeton University, Princeton, NJ 08544, USA
[*]  Correspondence: francois.gallaire@epfl.ch

**Abstract:** We perform simulations to study the hydrodynamics of a conical-shaped swimming micro-robot that ejects catalytically produced bubbles from its inside. We underline the nontrivial dependency of the swimming velocity on the bubble deformability and on the geometry of the swimmer. We identify three distinct phases during the bubble evolution: immediately after nucleation the bubble is spherical and its inflation barely affects the swimming speed; then the bubble starts to deform due to the confinement gradient generating a force that propels the swimmer; while in the last phase, the bubble exits the cone, resulting in an increase in the swimmer velocity. Our results shed light on the fundamental hydrodynamics of the propulsion of catalytic conical swimmers and may help to improve the efficiency of these micro-machines.

**Keywords:** catalytic microswimmers; bubble-propelled microswimmers; microrockets; numerical simulations; self-propulsion

## 1. Introduction

Microrockets, or conical microswimmers, are emerging as one of the most promising micro-swimmers. Their high swimming velocity, up to 50 body lengths per second, positions them among the fastest microswimmers [1–3]. Furthermore, their structure enables them to easily carry micro-objects, usually on the external cone surface [4,5]. These capabilities make microrockets attractive for several engineering applications, for example drug delivery [6–9]. The delivery of micro-particles has been realized either by functionalization of the external part of the cone surface [6] or by mechanically carrying particles at one end of the cone [10], while the first in vivo use of this technology for drug delivery has also been conducted [11].

The propulsion of conical swimmers is obtained via a catalytic decomposition, transforming chemical energy into propulsion energy. These micro-swimmers are composed of a cone, internally coated with a catalyst. When immersed in fuel, usually hydrogen peroxide, the active part starts to catalytically decompose the hydrogen peroxide into water and molecular oxygen $2H_2O_2 \rightarrow 2H_2O + O_2$ [12]. As a result of the catalytic decomposition, oxygen bubbles are generated and their growth inside the conical body propels the microswimmer.

In this study, we focus on the motion of a single conical microswimmer. In fact, despite the vast amount of experimental studies, their fundamental propulsion mechanisms remain quite unclear. An important aspect is the modeling of the micro-swimmer motion, previous studies concentrate either on the movement of the bubble when still inside the cone [13,14], or they consider mainly the locomotion due to the bubble exiting the cone [15,16]. Therefore, we proceed by solving numerically

the fluid motion generated by a deformable bubble inflating inside the conical microswimmer, from its nucleation until when it exits the cone.

We find three different phases of bubble growth with different characteristics, when the bubble is almost spherical inside the cone, when it grows and undergoes geometrical confinement and when it exits the cone, restoring its spherical shape. Finally, we show how these three different phases contribute to the cone displacement and we identify optimal geometrical parameters which maximize the cone velocity.

## 2. Problem Setup

We consider a micro-rocket consisting of a truncated hollow cone encapsulating an inflating deformable bubble of volume $V_B(t)$. The geometry of the micro-rocket is characterized by the inner radius $R$ of front opening, the contour-length $L$ of its shell and the opening angle $\theta$ between the shell and the $z$-axis of the cone, see Figure 1. We fix the aspect ratio $\xi = L/R = 10$, a design parameter relevant with the experiments [4,13,15].

To model the growing bubble inside the cone due to the emitted gas from the chemical reaction on the inner surface, we assume that all the reaction-emitted gas goes instantaneously into the bubble with a steady flux of $\dot{n} = \mathcal{A}\xi R^2$, where $\mathcal{A}$ denotes the molar flux per unit area, indicating the surface chemical reaction rate, and typically of order $10^2$ mol·m$^{-2}$·s$^{-1}$ [13]. This assumption is justified by the diffusion-dominated nature of the gas exchange [17], even if it neglects the escape of oxygen from the front or rear inlet of the cone. We argue that this simplifying hypothesis allows us to estimate the order of magnitude of the oxygen flux entering the bubble and might still capture the phenomenon qualitatively.

Because of the small size and low swimming speed of the micro-rockets, the inertia and mass effects can be safely neglected: the Reynolds Re $= \rho \bar{U} R / \mu$ is of the order $\mathcal{O}(10^{-4})$ if we use the experimentally measured typical average cone velocity $\bar{U} = 10^{-4}$ m/s and radius $R = 10^{-6}$ m [13], with $\rho$ and $\mu$ the fluid density and viscosity respectively. We hence solve the governing equations for the steady creeping flow. In addition, the Bond number is of the order $\mathcal{O}(10^{-6})$, hence the bubble only moves along the axis of the motor, and the problem is assumed to be axisymmetric. The viscosity of the bubble is negligible compared to that of the suspending fluid. A constant surface tension coefficient of the clean air-water interface $\gamma = 7.2 \times 10^{-2}$ N/m is adopted, neglecting soluto-capillary Marangoni effects. In this study, we will focus on the influence of the opening angle $\theta$ of the cone and the growth rate $\dot{n}$ of the bubble.

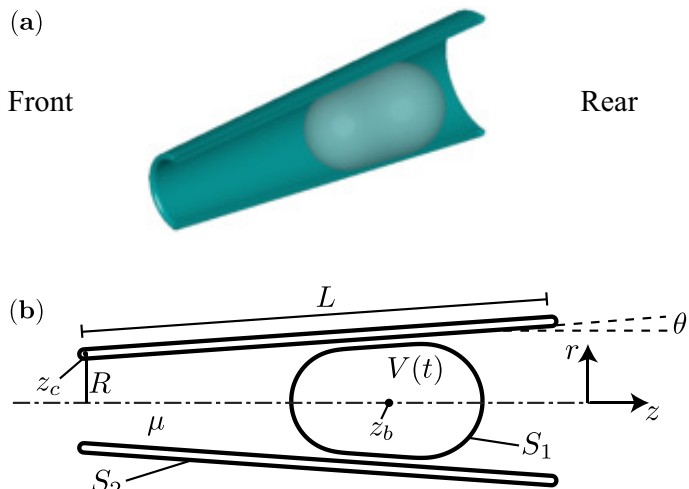

**Figure 1.** Sketch of the geometrical configuration: (**a**) 3D view of a section of the conical swimmer and (**b**) the corresponding side view.

### 2.1. Governing Equations

We solve the steady Stokes equations governing the fluid motion. By using $R$, $\frac{\gamma}{\mu}$, $\frac{R\mu}{\gamma}$ and $\frac{\gamma}{R}$ as the characteristic length, velocity, time and pressure scale, we obtain the dimensionless Stokes equations for the fluid velocity $\mathbf{u}$ and pressure $p$,

$$-\nabla p + \nabla^2 \mathbf{u} = 0, \quad \nabla \cdot \mathbf{u} = 0. \tag{1}$$

The cone translates with velocity $\mathbf{U}_c = \dot{z}_c \mathbf{e}_z$, thus the boundary condition (BC) on its surface $S_2$ is

$$\mathbf{u} = \mathbf{U}_c, \quad \text{for} \quad \mathbf{x} \in S_2. \tag{2}$$

On the bubble surface $S_1$, we impose the discontinuity of normal stress due to the surface tension force and disjoining pressure $\mathbf{\Pi}$,

$$\sigma \cdot \mathbf{n} - p_B \mathbf{n} = (\nabla_s \cdot \mathbf{n})\mathbf{n} + \mathbf{\Pi}(\mathbf{x}) \quad \text{for} \quad \mathbf{x} \in S_1, \tag{3}$$

where $\sigma$ is the stress tensor, $p_B$ the gas pressure inside the bubble, $\mathbf{n}$ the normal vector pointing into the suspending fluid and $\nabla_s = (\mathbf{I} - \mathbf{nn}) \cdot \nabla$ is the surface gradient operator. We model the disjoining pressure at the location $\mathbf{x} \in S_1$ of the bubble interface due to the proximity to the cone surface following [18], namely

$$\mathbf{\Pi}(\mathbf{x}) = \begin{cases} \int_{S_2} B \dfrac{e^{\delta - d} - 1}{e^\delta - 1} \dfrac{\mathbf{x} - \mathbf{y}}{d} dS(\mathbf{y}) & \text{for} \quad d \leq \delta, \\ 0 & \text{for} \quad d > \delta, \end{cases} \tag{4}$$

where $\mathbf{y}$ is a point lying on the cone surface, $d = |\mathbf{x} - \mathbf{y}|$, $\delta$ and $B$ denote the range and the magnitude of the force, respectively. This law mimics the repulsion due to an electric double layer potential [19]. The bubble volume $V_B$ is imposed as a global constraint and varies following the ideal gas law (variables indicated by $\sim$ denote dimensional quantities)

$$\begin{aligned} \frac{d(\tilde{p}_B \tilde{V}_B)}{d\tilde{t}} &= \dot{n} \mathcal{R} T_0 \\ &= \mathcal{A} \xi R^2 \mathcal{R} T_0, \end{aligned} \tag{5}$$

where $\mathcal{R}$ is the ideal gas constant and $T_0 = 300$ K is the ambient room temperature. When imposing the molar flux, the volume increase is found by assuming that the internal bubble pressure is the same as that of a spherical bubble of the same volume

$$\tilde{p}_B = \tilde{p}_0 + \frac{2\gamma}{\left(\frac{3}{4\pi}\tilde{V}_B\right)^{1/3}}. \tag{6}$$

Substituting Equation (6) into (5) followed by non-dimensionalization, we obtain

$$\frac{dV_B}{dt} = \frac{\text{Ca}}{\beta + C V_B^{-1/3}}, \tag{7}$$

where $\text{Ca} = \frac{\mathcal{A}\xi R \mathcal{R} T_0}{\gamma} \frac{\mu}{\gamma}$ represents the capillary number, $\beta = \frac{p_0 R}{\gamma}$ and $C = \frac{4}{3(\frac{3}{4\pi})^{1/3}}$. Ca indicates the ratio between the reaction-induced bubble inflation velocity $\frac{\mathcal{A}\xi R \mathcal{R} T_0}{\gamma}$ over the capillary velocity $\gamma/\mu$. The cone velocity $\mathbf{U}_c$ is obtained by imposing the global force-free condition on the system, implying that the net force on the bubble due to the repulsive forces is applied back to the cone,

$$\int_{S_2} \mathbf{f} \cdot \mathbf{e}_z \mathrm{d}S + \int_{S_1} \mathbf{\Pi} \cdot \mathbf{e}_z \mathrm{d}S = 0. \tag{8}$$

By solving Equations (1)–(8) numerically (see Section 2.2), we get the instantaneousness cone velocity $\mathbf{U}_c$ and the bubble interface velocity $\mathbf{u}$. The time evolution of the system is obtained by integrating

$$\frac{\mathrm{d}z_c}{\mathrm{d}t} = \mathbf{U}_c \cdot \mathbf{e}_z, \tag{9}$$

$$\frac{\mathrm{d}\mathbf{x}}{\mathrm{d}t} = \mathbf{u} \quad \text{for} \quad \mathbf{x} \in S_1, \tag{10}$$

in time. The initial bubble radius is $R_0 = 0.5$ and its center is located at $z_0\mathbf{e}_z$, with $z_0 = 5$. These assumptions allow us to avoid the modeling of bubble nucleation and its initial motion when the bubble typical length scale is much smaller compared to the cone length scale, this seems to be reasonable as very small bubbles are not expected to influence the cone motion greatly. See [20,21] for a more detailed analysis of bubbles nucleation on micro-rockets surfaces and their motion at short times, respectively. The capillary number is approximately $Ca \in [10^{-5}, 10^{-4}]$, at most $Ca \in [10^{-4}, 10^{-3}]$ in presence of surfactants, as commonly used in experiments to stabilize the bubble interface. Nevertheless, larger values $Ca \in [10^{-3}, 5 \times 10^{-1}]$ are adopted in our study because the computation becomes prohibitively expensive for the realistic Ca values owing to strong numerical stiffness. With this limitation in mind, we argue that this might not compromise our goal of understanding the physical process qualitatively. We fix the intensity of the disjoining pressure to $B = 10$, activation distance $\delta = 0.2$, and $\beta = 1$ as in [17]. These values are larger than the physical ones in order to ease our numerical simulations which would become prohibitively time-consuming when the lubrication films becomes increasingly thin.

### 2.2. Numerical Methods

We use the axisymmetric boundary integral method (BIM) to solve the flow [22]. The BIM equations are integrated along the contour line $l_1$ and $l_2$ in the meridional plane, representing the surface $S_1$ and $S_2$ respectively. The velocity on a point $\mathbf{x}_0$ lying on the bubble interface is

$$4\pi\mathbf{u}(\mathbf{x}_0) = -\int_{l_1} \mathbf{M}(\mathbf{x}_0, \mathbf{x}) \cdot \Delta\mathbf{f}(\mathbf{x})\mathrm{d}l + \int_{l_1}^{\mathrm{PV}} \mathbf{n}(\mathbf{x}) \cdot \mathbf{q}(\mathbf{x}_0, \mathbf{x}) \cdot \mathbf{u}(\mathbf{x})\mathrm{d}l - \\ \int_{l_2} \mathbf{M}(\mathbf{x}_0, \mathbf{x}) \cdot \mathbf{f}(\mathbf{x})\mathrm{d}l + \mathbf{n}(\mathbf{x}_0)\left(\int_{l_1} \mathbf{u}(\mathbf{x}) \cdot \mathbf{n}(\mathbf{x})\mathrm{d}l - \dot{V}_B\right), \tag{11}$$

where $\Delta\mathbf{f} = \sigma \cdot \mathbf{n} - p_B\mathbf{n}$. When $\mathbf{x}_0$ lies on the cone wall,

$$8\pi\dot{z}_c\mathbf{e}_z = -\int_{l_1} \mathbf{M}(\mathbf{x}_0, \mathbf{x}) \cdot \Delta\mathbf{f}(\mathbf{x})\mathrm{d}l + \int_{l_1} \mathbf{n}(\mathbf{x}) \cdot \mathbf{q}(\mathbf{x}_0, \mathbf{x}) \cdot \mathbf{u}(\mathbf{x})\mathrm{d}l - \\ \int_{l_2} \mathbf{M}(\mathbf{x}_0, \mathbf{x}) \cdot \mathbf{f}(\mathbf{x})\mathrm{d}l + \mathbf{n}(\mathbf{x}_0)\left(\int_{l_1} \mathbf{u}(\mathbf{x}) \cdot \mathbf{n}(\mathbf{x})\mathrm{d}l - \dot{V}_B\right). \tag{12}$$

More specifically, $\mathbf{M}$ and $\mathbf{q}$ are the Green's functions of the Stokes equations forced by a ring of point forces acting in $\mathbf{x}$ after azimuthal integration, as explained in detail in Refs. [22,23]. The third term on the right hand side of Equations (11) and (12) requires special treatment in order to avoid that the stress $\mathbf{f}(\mathbf{x})$ is defined up to a constant. Mathematically, this is equivalent to eliminating a neutral mode in the integral operator, as implemented in [23]. The last term on the right hand side accounts for the bubble inflation [24]. The force-free condition (8) writes as

$$\int_{l_2} r\,\mathbf{f} \cdot \mathbf{e}_z \mathrm{d}l + \int_{l_1} r\,\mathbf{\Pi} \cdot \mathbf{e}_z \mathrm{d}l = 0. \tag{13}$$

The cone geometry is discretized into $N_w$ straight constant ($P_0$) elements and the bubble into $N_b$ curved linear ($P_1$) elements. The discretization of Equations (11)–(13) results in a linear system of size ($2N_w + 2N_b + 3$). Its numerical solution results in the stresses on the cone surface **f**, the interface velocity **u** and the cone velocity $\dot{z}_c$. Converged results are reached by choosing $N_w = 124$ and $N_b = 20$ for a bubble of initial radius $R_0 = 0.5$. Moreover, adaptive mesh refinement is performed when necessary to capture the evolution of thin lubrication film with an adequate numerical accuracy. A second order Runge-Kutta scheme is used for time-marching the system.

## 3. Results

### 3.1. Velocity Field and Micro-Rocket Velocity over One Bubble Cycle

A typical bubble evolution and associated cone motion is depicted in Figure 2 for Ca = 0.01 and $\theta = 1°$. The cycle starts when the bubble is inside the micro-rocket and ends when it exits, recovering a spherical shape (see Figure 2a–d). Although we realize that multiple bubbles might appear simultaneously in the cone in the experiments, we hereby consider only a single bubble to simplify our model. This simplification enables us to better focus on the fundamental propulsion mechanism at play. Intuitively, since the bubble tends to recover a spherical shape to minimize its dimensionless surface energy, we can argue that a deformed bubble is storing energy that will be released for propulsion later on. This motivates us to monitor the excess surface area $\Delta A = A - A_0$, where $A_0$ denotes the surface area of a spherical bubble. This scalar quantifies the surface energy: $\Delta A$ is zero for spherical bubbles and larger than zero for deformed ones. The time evolution of $\Delta A$ is featured by the following three phases:

- Phase *I* (spherical phase): the unconfined bubble is spherical, from $t = 0$ to $t \approx 700$. As shown in Figure 2a, the fluid is expelled from both openings of the cone, due to the inflation of the bubble. $\Delta A$ is almost equal to zero during this phase, and both the cone and bubble move slowly.
- Phase *II* (migration phase): the bubble is confined and squeezed inside the cone, from $t \approx 700$ to $t \approx 2100$. As shown in Figure 2b,c, the bubble starts to translate fast due to the geometrical confinement, drawing fluid from the front opening. During this phase, the bubble becomes more squeezed, leading to an increasing $\Delta A$ (see Figure 2e). Both the cone and bubble speed up (see Figure 2f,g) due to the increasing confinement.
- Phase *III* (recoil phase): the bubble exits the cone and rapidly recovers its spherical shape, when $t \approx 2100$ to $t \approx 2600$ (see Figure 2d). This phase clearly starts when $\Delta A$ reaches its maximum, continuing when $\Delta A$ decreases. During this phase, the rapid release of energy due to bubble relaxation leads to the maximum velocities of the cone and bubble (i.e., maximum slope shown in Figure 2f,g).

The cone velocity evolution over one bubble cycle seems to provide an explanation on the unsteady movement of the micro-rockets observed experimentally [13]. Namely, the cone translates relatively slowly during phase *I* and *II*, while it accelerates during phase *III* resulting from the sudden release of surface energy. Figure 3 depicts the velocity field induced by a moving cone (a), bubble (b) and the sum of the two (c) at $t = 1000$.

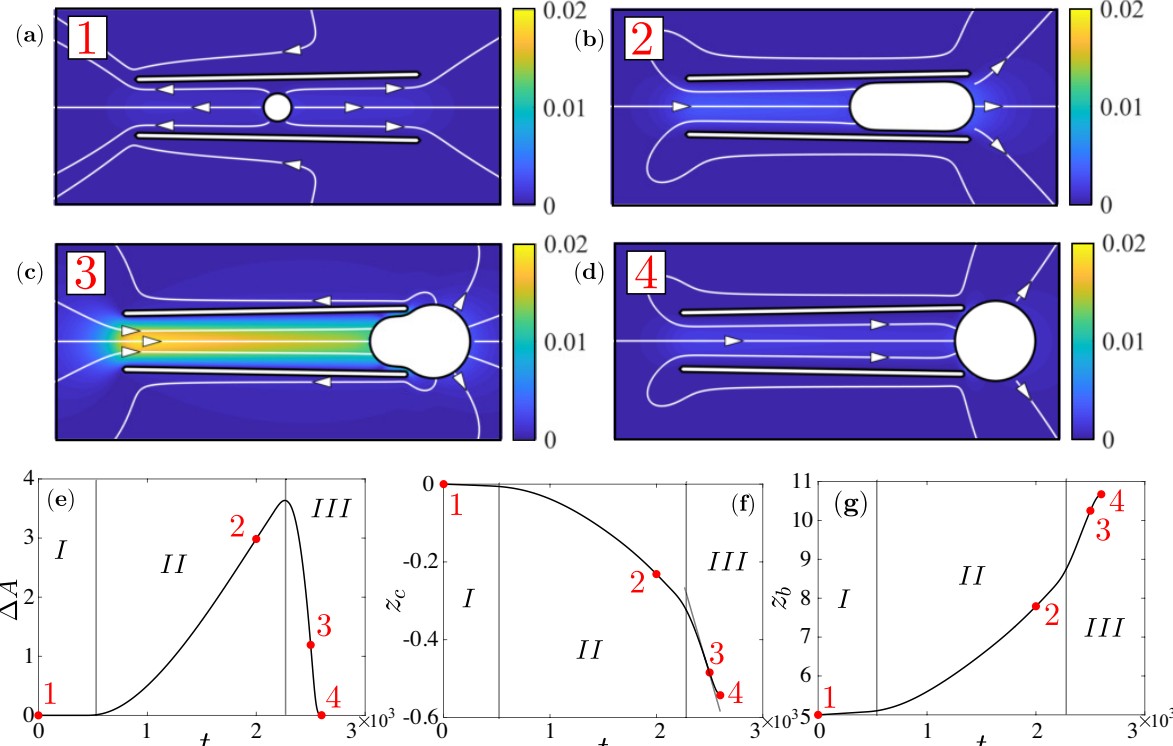

**Figure 2.** A bubble cycle for Ca = 0.01 and $\theta = 1°$. (**a**–**d**) velocity magnitude and streamlines in the lab frame at four times from the bubble nucleation until when the bubble exits the cone; (**e**) Excess surface area versus time; (**f**) Cone and (**g**) bubble displacements versus time. The vertical lines separate three phases.

The motion of the cone and the bubble contribute to the velocity field with a Stokeslet, decaying like $1/\rho$ [25], where $\rho = \sqrt{z^2 + r^2}$, because they are not force-free due to the disjoining pressure (see Figure 3a,c,d,f). However, the micro-rocket is globally force free due to Equation (13), leading to a $1/\rho^2$ stresslet-like decay of the velocity field (see Figure 3c–f).

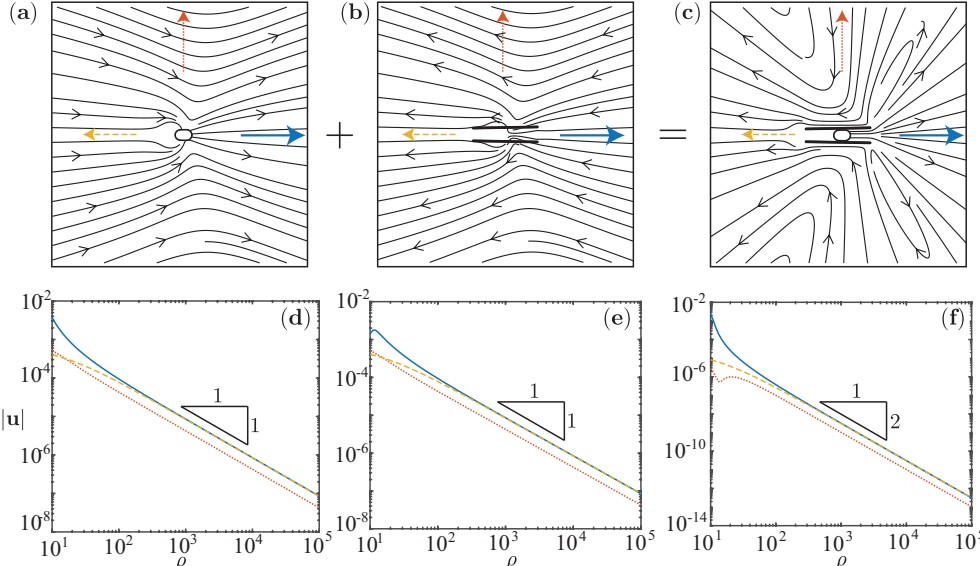

**Figure 3.** Far field streamlines and velocity magnitude at $t = 1000$ for Ca = 0.01 and $\theta = 1°$. (**a**–**c**) Cone, bubble and cone-bubble induced velocity field respectively; (**d**–**f**) decay of velocity in three directions, in the rear (yellow) and forward (blue) direction of the cone, and in the radial (orange) direction.

### 3.2. Influence of the Opening Angle θ

The opening angle $\theta$ of the cone is an important design parameter. In this section, we examine its effect on the average micro-rocket velocity $\bar{U}$. We fix Ca $= 0.01$ and vary $\theta$, Figure 4a shows the time evolution for three opening angles, $\theta = 0.25°$, $1°$ and $2°$. We observe that the duration $\Delta t$ of the bubble cycle for $\theta = 1°$ is smaller than those of the other angles, indicating a non-monotonic variation of $\Delta t$ in $\theta$. First, we notice that the excess surface area at the beginning of the recoiling phase is larger at a smaller $\theta$, because the geometrical confinement is larger for smaller $\theta$ (see Figure 4b). Therefore, while recoiling, the bubble releases more energy, leading to a higher fluid velocity (in fact, both bubble and cone displace faster, see Figure 4c,d). Therefore, it seems that this non-monotonic behavior results from phase $I$ and $II$, as suggested by the non-monotonic starting point of the recoiling phase shown in Figure 4b.

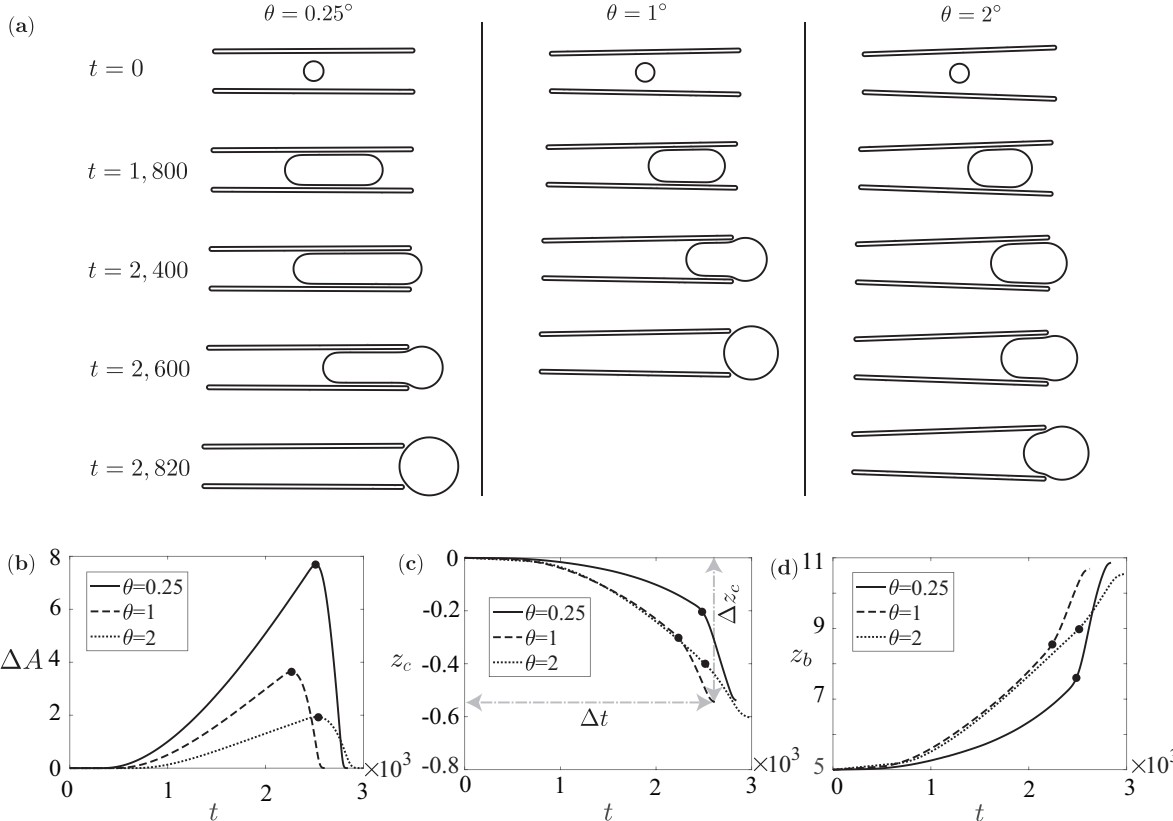

**Figure 4.** (**a**) Snapshots of the bubble-cone configuration for three opening angle $\theta = [0.25, 1, 2]°$, when Ca $= 0.01$, and corresponding time evolution of (**b**) excess surface area $\Delta A$; (**c**) cone displacement $z_c$ and (**d**) bubble displacement $z_b$. The dots indicate the times when the maximum excess surface area is reached.

In Figure 5, we scrutinize the source of the non-monotonicity. First of all, the average velocity $\bar{U} = \Delta z_c / \Delta t$ shows an optimum at $\theta \approx 1°$. This optimum is attributed to the minimum time $\Delta t$, because $\Delta z_c$ increases almost monotonically as shown in Figure 5a,b. More specifically, the minimum $\Delta t$ results from the minimum $\Delta t_{I+II}$ (the duration of phase $I$ and $II$), whereas $\Delta t_{III}$ (the duration of phase $III$) increases monotonically. Therefore, the non-monotonic average velocity, with a maximum for $\theta \approx 1°$ shown in Figure 5c, is dictated almost exclusively by phase $I$ and $II$. In fact, while the third phase exhibits a monotonic decrease of the average velocity $\bar{U}_{III} = \Delta z_c^{III} / \Delta t_{III}$ in $\theta$ ($\Delta t_{III}$ monotonically increases and $\Delta z_c^{III}$ monotonically decreases), the first two phases show an optimum for $\bar{U}_{I+II} = \Delta z_c^{I+II} / \Delta t_{I+II}$. We will attempt using a model to explain this non-trivial optimum in Section 3.3.

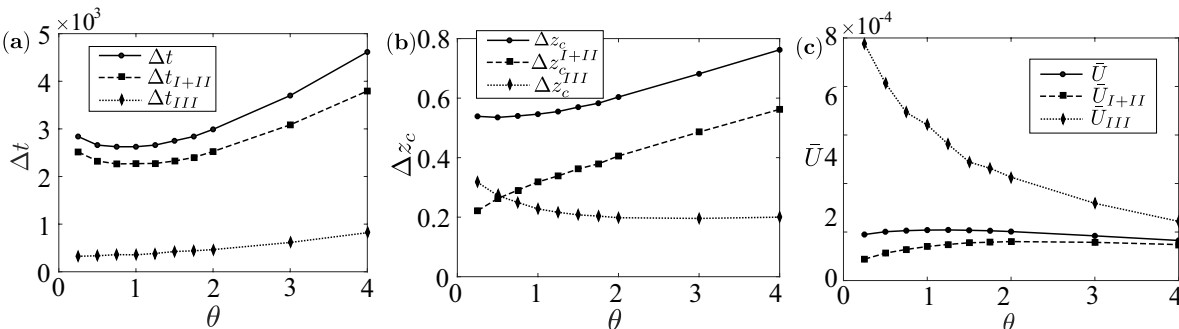

**Figure 5.** Relevance of the two phases over one bubble cycle: (**a**) Bubble cycle duration $\Delta t$ and duration of first and second phase $\Delta t_{I+II}$ and third phase $\Delta t_{III}$, respectively; (**b**) Cone displacement over one bubble cycle $\Delta z_c$, cone displacement over the first and second phase $\Delta z_c^{I+II}$ and third phase $\Delta z_c^{III}$; (**c**) Micro-rocket average velocity over one bubble cycle $\bar{U}$, average velocity over the first and second phase $\bar{U}_{I+II}$ and third phase $\bar{U}_{III}$.

We plot in Figure 6 the average velocity versus the opening angle for different capillary numbers. The optimum existing for all capillary numbers considered is attained approximately for $\theta \approx 1°$. Interestingly, the average velocity does not vary significantly (at least keeping $\theta < 4°$), pointing to the idea that, when increasing $\theta$, a reduced performance of the second phase is counterbalanced by an improved performance of the first phase to certain extent. This might explain the robustness in the micro-rocket swimming velocity observed in experiments, regardless of the different opening angles due to different fabrication techniques.

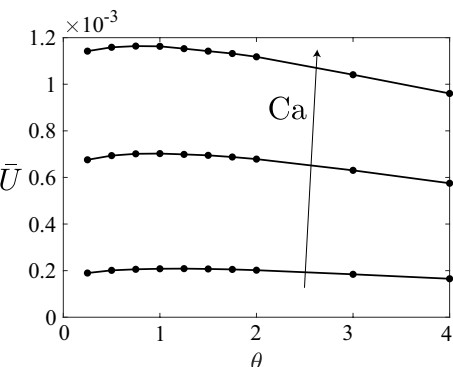

**Figure 6.** Average velocity of the micro-rocket versus the cone angle $\theta$ for Ca $= [0.01, 0.05, 0.1]$.

### 3.3. An Empirical Spring-Like Model for Confined Bubbles

We hereby aim to unravel the existence of the optimal angle for the swimming velocity. We focus on the modeling of the phase *II*, where the optimum most likely originates, while phase *I* is probably less important because it seems to depend only weakly on $\theta$. We proceed empirically, by drawing an analogy between a simple spring, where its restoring force is proportional to the displacement due to Hooke's law $F \propto \Delta x$ [26], and a bubble where we argue that the restoring force is proportional to the excess area $F \propto \Delta A$ (see Figure 7a). By assuming this linear behavior we are neglecting the possible dependence of the force upon other parameters, such as the bubble volume, and the hydrodynamics, such as the lubrication force scalings due to the thin film; however, this simplified model seems to be sufficient to explain the maximum velocity from phase *II*. Assuming small $\theta$, the force acting on the cone wall in the axial direction due to the restoring force of the bubble (see Figure 7b) writes

$$\frac{F_z}{\theta} \propto \Delta A \rightarrow F_z \propto \Delta A\, \theta. \tag{14}$$

For a given volume, the excess area decreases when the opening angle increases because of the decreasing geometrical confinement. In Figure 7d, we compute numerically the excess area assuming that the bubble is composed of a cone section closed by two semi-spherical caps whose center of mass is placed in the middle of the cone. When the volume and film thickness are fixed, the bubble shape is uniquely determined. For small $\theta$, the excess surface area seems to decrease exponentially, although we have not yet attempted to show it analytically. Therefore, $F_z$ shows a non-monotonic variation in $\theta$. Namely, $F_z(0) = 0$ because the projection of the force in the axial direction is zero, while it is small for sufficiently large $\theta$ because the exponential decrease of $\Delta A$ dominates over the linear increase of the projection angle (see Figure 7e). Thus, assuming that the micro-rocket velocity scales linearly with $F_z$ and that its drag coefficient does not vary with $\theta$ when it is small enough, these two effects explain the non-monotonic behavior of the micro-rocket velocity observed in Section 3.2.

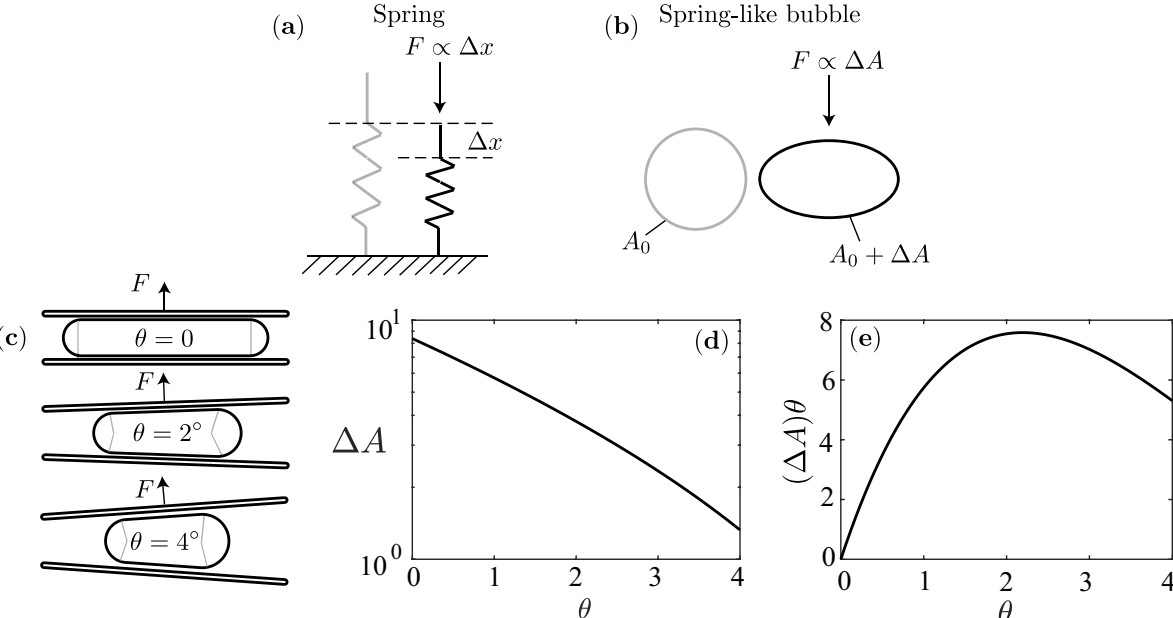

**Figure 7.** Spring-bubble empirical model. (**a**) Sketch of Hooke's law; (**b**) Sketch of the analogy to Hooke's law for bubbles where $F \propto \Delta A$; (**c**) Bubble of volume $V_0$ squeezed in cones of different opening angles; (**d**) Excess area $\Delta A$ versus $\theta$ assuming that the bubble is composed of a conical section and two semi-spherical caps; (**e**) $F_z \propto (\Delta A)\theta$ versus $\theta$.

### 3.4. Influence of the Capillary Number Ca

We vary the capillary number Ca to investigate the impact of chemical reaction rate $\mathcal{A}$ on the performance of the micro-rockets. In fact, several studies have focused on employing enhanced catalysts to maximize the swimming velocity [27,28].

In Figure 8a, we show the results for different Ca values. When increasing Ca, the bubble inflates more rapidly. Consequently, the bubble cycle is shorter and the cone displacement is larger; both effects increase the average velocity $\bar{U}$ with Ca (see Figure 8b), and we seemingly identify a scaling relation $\bar{U} \sim \text{Ca}^{0.75}$. Varying the opening angle does not seem to change the scaling. As a consequence, the change in the chemical flux (leading to a Ca variation) will not directly result in a proportional variation of the swimming velocity due to the slight sublinear scaling. We have not yet been able to explain the physical origin of this scaling.

Figure 9 depicts the phases duration and corresponding displacement versus Ca. We observe that the duration of the first and second phase decreases when increasing Ca. In particular, the first phase scales roughly as the inverse of Ca, which is expected because it corresponds to the inverse of the inflation rate. The duration of the third phase is roughly Ca independent. The displacement attained during the first phase is constant because it is given by an inflating quasi spherical bubble

and therefore is independent on the inflation rate, as shown in [17] for spherical bubbles. Interestingly, also the displacement during the second phase remains almost constant. The displacement during the third phase increases with Ca. In fact, when Ca increases, the bubble squeezes more inside the cone, thus storing more surface energy that is released during relaxation. All together, these different phases lead to the ad-hoc velocity scaling shown in Figure 8.

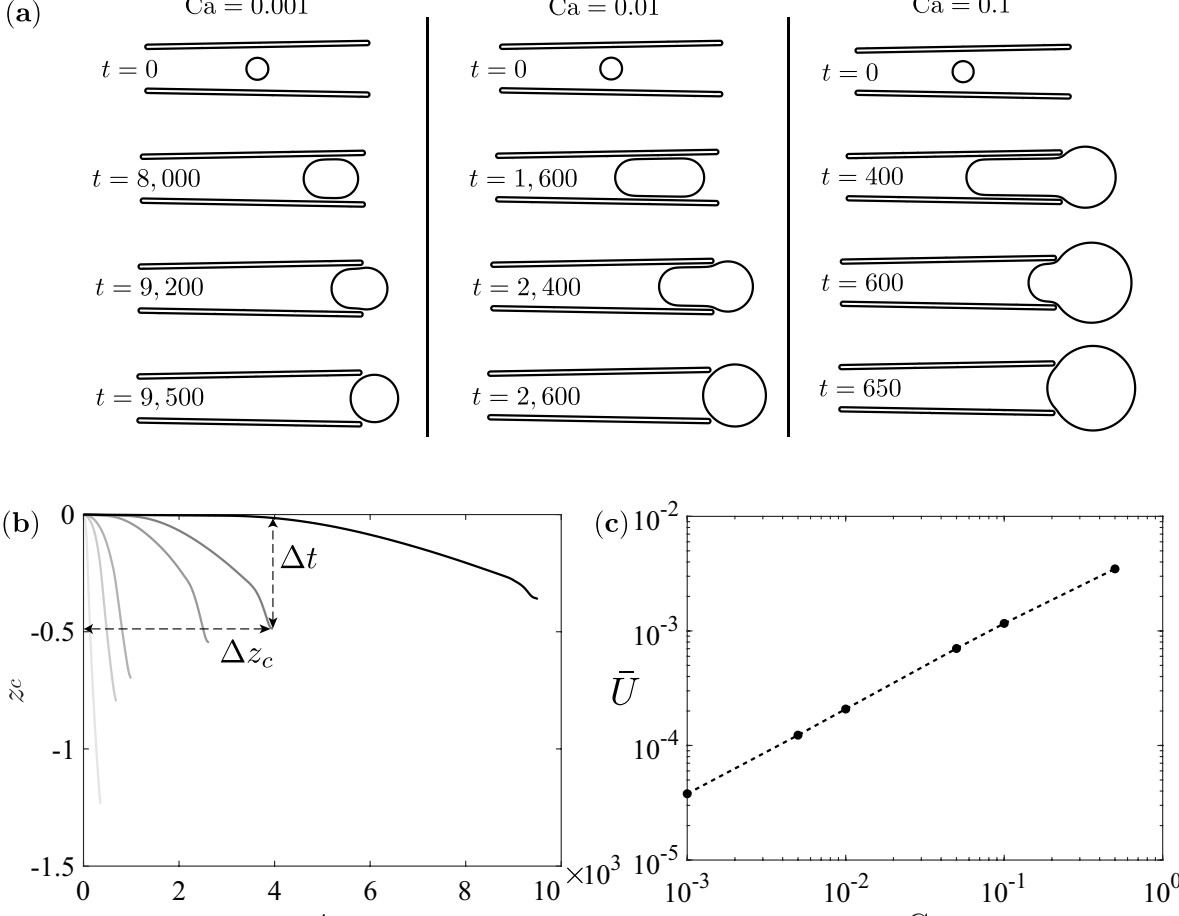

**Figure 8.** (**a**) Snapshots for the time evolution at different capillary numbers, $\theta = 1°$; (**b**) Cone position versus time, black to light grey line for increasing capillary numbers Ca $= [0.001, 0.005, 0.01, 0.05, 0.1, 0.5]$; (**c**) Average velocity versus capillary number.

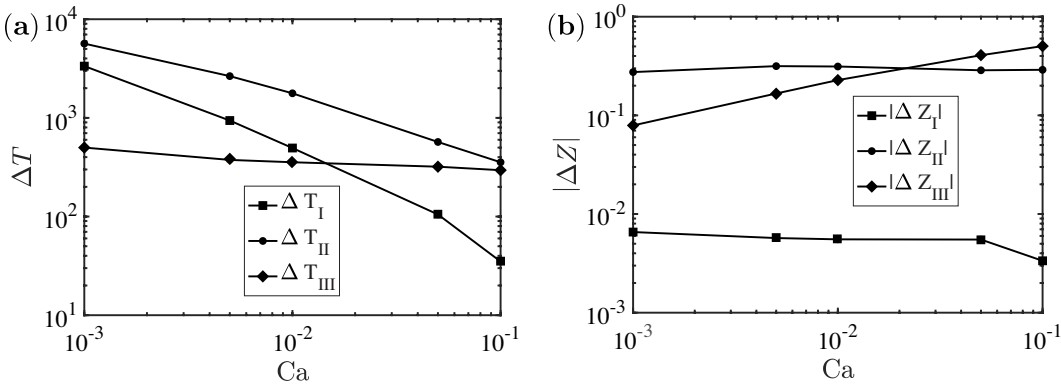

**Figure 9.** (**a**) Phase duration and (**b**) corresponding displacement attained versus capillary number.

### 3.5. Critical Threshold for Sustained Bubble Ejection

In this section we study the influence of the initial bubble position on the bubble cycle. Fixing Ca $= 0.1$, we examine two cases with $z_b(0) = 4$ and $z_b(0) = 2$. When $z_b(0) = 4$ the bubble cycle is similar to what is observed ($z_b(0) = 5$ in all previous simulations, corresponding to the middle of the cone), but when $z_b(0) = 2$ the bubble exits from the front opening, as shown in Figure 10. This situation might correspond to a situation observed experimentally, where the bubble exits from both the rear and front openings, spoiling a robust unidirectional motion [15,29]. This mechanism might even be exploited to achieve complex bidirectional motions, as described in [30] as an alternative strategy to the ultrasound control [31]. In fact, when the bubble exits from the front opening, the micro-rocket displaces from left to right while it displaces from right to left when the bubble exits from the rear opening. For a smaller capillary number, this problem is less evident because the bubble deforms less and soon starts to translate towards the rear opening. Since the capillary number increases at higher chemical fluxes $\dot{n} = \mathcal{A}\xi$, using enhanced catalysts or longer micro-rockets might worsen this situation hence posing a serious challenge to achieve a sustained bubble cycle.

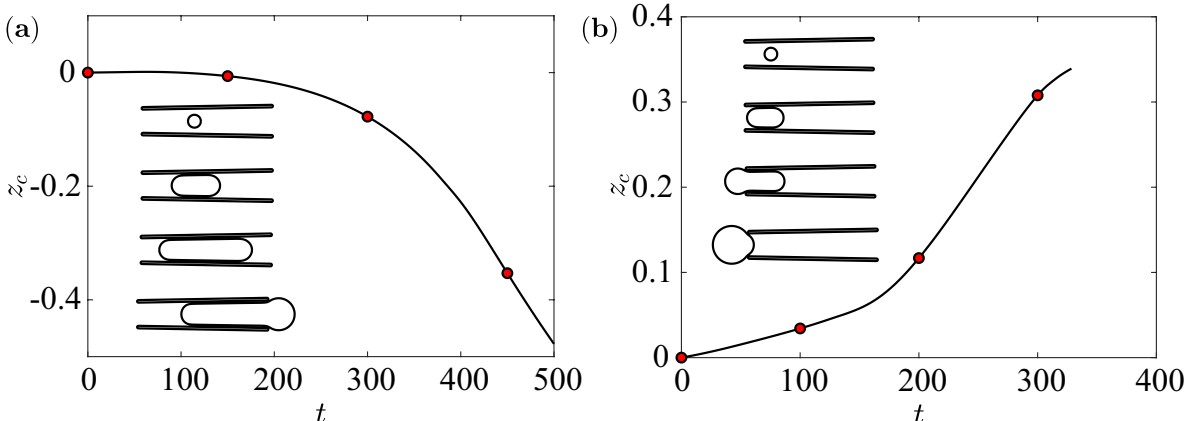

**Figure 10.** Cone position versus time when (**a**) $z_b(0) = 4$ and (**b**) $z_b(0) = 2$. The insets show snapshots corresponding to the circles.

## 4. Conclusions

We have investigated numerically the motion of a catalytic micro-rocket due to the inflation of a deformable bubble. We have identified a physical quantity, the excess surface area, which helps characterizing three phases of the bubble growth and corresponding micro-rocket motion. The first two phases correspond to low microrocket velocity, when the bubble is confined, while the third one results in a high micro-rocket velocity, when the bubble exits and recoils. By studying the influence of the different phases when varying the design parameters (as the opening angle of the cone), we have identified the optimal working conditions and concluded on their robustness for small opening angles. This leads to the conclusion that as long as the opening angle is small ($\theta < 4°$), there is no particular need of optimizing this parameter from the hydrodynamic point of view. This weak optimum is then physically interpreted with the help of an empirical model. Moreover, we have observed a sublinear scaling of the average swimming velocity with the capillary number. As a consequence, enhanced catalysts, which are extensively studied, would possibly lead to an increase of velocity but a decrease of efficiency. Finally, we have analyzed the dependence of the swimming velocity upon the initial bubble position: in particular, when a bubble is close to the front opening it might exit from there causing a micro-rocket motion in the opposite direction than expected. This phenomenon, that has been observed in the literature [15], is more likely to happen at high capillary number, which might pose technological problems using enhanced catalysts surfaces [27]. In future work, we plan to incorporate the chemical reaction model developed in [17].

**Author Contributions:** Conceptualization, G.G., L.Z. and F.G.; methodology, G.G. and L.Z.; software, G.G. and L.Z.; validation, G.G.; investigation, G.G., L.Z. and F.G.; writing—original draft preparation, G.G., L.Z. and F.G.; writing—review and editing, G.G., L.Z. and F.G.; visualization, G.G.; funding acquisition, F.G.; F.G., L.Z. and G.G. have contributed equally.

**Funding:** This project has received funding from the European Research Council (ERC) under grant agreement SimCoMiCs-280117 (G.G., L.Z., and F.G.) and from the Swedish Research Council under the Postdoc Grant agreement 2015-06334 (L.Z.). The computer time has been provided by SCITAS at EPFL.

**Conflicts of Interest:** The authors declare no conflict of interest.

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
