# Peer review of "The Hydrodynamics of a Micro-Rocket Propelled by a Deformable Bubble"

_fluids, doi:10.3390/fluids4010048_

Round 1

Reviewer 1 Report

Please see attached report.

Author Response

see attached pdf

Reviewer 2 Report

The study is rigorous and also provides another perspective on bubble-propelled locomotion. The manuscript is well-written and can be considered for publication. While single bubble evolution, confinement and ejection is an important point of study, the interactions between bubbles that result in locomotion is also an important factor. A comment on how bubbles interact within each other, will allow the wider readership and an insight of how inter-bubble interactions allow motion to take place. The simulations study should also associate itself with classical and recent experimental studies on bubble-propelled micromotors. A note is that of classical experimental study about evolution of microbubbles in such catalytic tubular micromotors is not cited. [Solevov et al., Catalytic microtubular jet engines self-propelled by accumulated gas bubbles] Recent studies on the stepwise or continuous motion of such micromotors for ejection of bubbles for locomotion should also be cited and the use of different-sized bubbles for motion. [Wang et al., From Nanomotors to Micromotors: The Influence of the Size of an Autonomous Bubble-Propelled Device upon Its Motion] The authors can also consider and cite the study of both the use of simulation and experimental work on the use of catalytically ejected bubbles for innovative ways of propulsion. [Moo et al., Bjerknes Forces in Motion: Long‐Range Translational Motion and Chiral Directionality Switching in Bubble‐Propelled Micromotors via an Ultrasonic Pathway] A few grammar errors and change in title of the parts. Line 35, 115 and 140 It should be exits and not exists Page 11 227 Is this discussion or conclusion? Page 11 230 characterizing

Author Response

see attached pdf

Reviewer 3 Report

The present manucript presents a theoretical study over the hydrodynamics of bubble evolution in catalytic micromotors moving by bubble-recoil mechanism, being unique for considering for the first time the deformation of such bubble inside its asymmetric body. Although the work is well structured, and the work is of interest for the micromotor's community for simulating an effect that is obvious from the experimental works already reported (bubble deformation), some specific points should considered before publication,

1) In the introduction, authors mention that transport of certain micro-objects is generally held on the outer surface of the motor' body. This is indeed not true, as tubular micromotor are the only kind of motors able to perform pick-up events by sucking the cell from one of its ends while it's moving (Sanchez S. et al., Chem. Commun. 2011, 47, 698). Such example should be included, as well as specifying that the micro-objects capture and transport is possible by chemical functionalizing the microrobot' outer surface. 

2) Regarding the reported works on the bubble propulsion mechanism from a fundamental point of view, it is missing the work from Fomin et al. (IEEE T. Robot. 2014, 30, 40). 

3) Although in the manuscript is defined the opening angle range considered, it is not clearly specified for which size range (length) the current study works with/considers. That is important, as each fabrication approach (i.e. rolled-up technology, template-based, 3D printing) covers different shapes/angles, and sizes. 

4) In page 4, line 85, the authors mention "very small bubbles are not expected to influence the cone motion greatly". I desagree, specially because this is a factor that the same authors considered in their previous paper (Adv. Funct. Mater. 2018, 28, 1800686). Indeed, people generally uses surfactant to obtain smaller bubbles, which leads to higher speeds when at the right fuel concentration. In line with this comment, I understand that is complicated for the authors to consider the surfactant effect during the micromotor motion, but at least they should mention that they are aware of its key role during the micromotor motion. 

5) The authors should clarify the following sentence "As a consequence, a change in chemical flux will not result in the same variation of the swimming velocity due to the slight sublinearity of the scaling", as it is not easy to see a logical relation between the two sentences.

6) Page 11, line 222: "In fact, when the bubble exits from the front opening, the micro-rocket displaces from left to right". The authors should clarify in which experimenta lcase this has been observed (in conclusions they mention that in reference 11, but it is not clear to me, as the case that they claim that there was rear motion there was bubbling from both sides). 

Regarding future works, I suggest the authors mainly study the fluid motion and bubble nucleation, as well as how this event affects to their resulting motion, but I suggest them to consider the study of the local mixing/microvortices that has extensively been reported in the field from an experimental point of view by working with passive particles as tracers and observing its behavior according to the motor' motion. 

Author Response

see attached pdf

Round 2

Reviewer 1 Report

The authors have thoroughly answered all questions from the three referees and revised the manuscript accordingly. In my view, it is now suitable for publication in its present form.